**Data Availability Statement:** All relevant data are within the manuscript.

**Funding:** The authors received no specific funding for this work.

# Diagnostics to support the control of scabies–Development of two target product profiles

**Michael Marks**[1,2,3]*, **Jodie McVernon**[4,5,6,7], **James S. McCarthy**[4], **Wendemagegn Enbiale**[8,9], **Christopher Hanna**[10], **Olivier Chosidow**[11,12,13], **Daniel Engelman**[14,15,16], **Kingsley Asiedu**[17], **Andrew Steer**[14,15,16]

**1** Department of Clinical Research, Faculty of Infectious and Tropical Diseases, London School of Hygiene and Tropical Medicine, London, United Kingdom, **2** Hospital for Tropical Diseases, University College London Hospital, London, United Kingdom, **3** Division of Infection and Immunity, University College London, London, United Kingdom, **4** Department of Infectious Diseases, The University of Melbourne at the Peter Doherty Institute for Infection and Immunity, Melbourne, Australia, **5** Victorian Infectious Diseases Laboratory Epidemiology Unit, Royal Melbourne Hospital at The Peter Doherty Institute for Infection and Immunity, Melbourne, Australia, **6** Centre for Epidemiology and Biostatistics, Melbourne School of Population and Global Health, The University of Melbourne, Melbourne, Australia, **7** Infection and Immunity Theme, Murdoch Childrens Research Institute, Melbourne, Australia, **8** Bahir Dar University, college of Medicine and Health sciences, Dermatovenerology Department, Bahir Dar, Ethiopia, **9** Amsterdam UMC, University of Amsterdam, Department of Dermatology, Amsterdam Institute for Infection and Immunity (AII), Academic Medical Centre, Amsterdam, Netherlands, **10** Global Project Partners, Oakland, California, United States of America, **11** AP-HP, Hôpitaux universitaires Henri Mondor, Service de Dermatologie, Créteil, France, **12** Research group Dynamyc, EA7380, Faculté de Santé de Créteil, Ecole nationale vétérinaire d'Alfort, USC ANSES, Université Paris-Est Créteil, Créteil, France, **13** GrIDIST, Groupe Infectiologie Dermatologique–Infections Sexuellement Transmissibles, Société Française de Dermatologie, Paris, France, **14** Tropical Diseases, Murdoch Children's Research Institute, Melbourne, Australia, **15** Department of Paediatrics, University of Melbourne, Melbourne, Australia, **16** Melbourne Children's Global Health, Royal Children's Hospital, Melbourne, Australia, **17** Department for the Control of Neglected Tropical Diseases, World Health Organization, Geneva, Switzerland

* michael.marks@lshtm.ac.uk

## Abstract

### Background

Scabies was added to the WHO NTD portfolio in 2017 and targets for the control of scabies were included in the 2021–2030 WHO NTD roadmap. A major component of scabies control efforts a strategy based on mass drug administration (MDA) with ivermectin. Currently diagnosis of scabies relies on clinical examination with a limited role for diagnostic testing. Under the recommendation of the WHO Diagnostic Technical Advisory Group (DTAG) for Neglected Tropical Diseases, a working group was assembled and tasked with agreeing on priority use cases for and developing target product profiles (TPPs) for new diagnostics tools for scabies.

### Methodology and principal findings

The working group convened three times and established two use cases: establishing if the 10% threshold for mass drug administration had been reached and if the 2% threshold for stopping mass drug administration has been achieved. One subgroup assessed the current diagnostic landscape for scabies and a second subgroup determined the test requirements

**Competing interests:** The authors have declared that no competing interests exist.

for both use cases. Draft TPPs were sent out for input from stakeholders and experts. Both TPPs considered the following parameters: product use, design, performance, configuration, cost, access and equity. The group considered the use of the tests as a single step process or as part of a two step process following initial clinical examination. When used a single step test (the ideal scenario) for starting MDA a new diagnostic required a sensitivity of ≥92% and a specificity of ≥98%. When used a single step test (the ideal scenario) for stopping MDA a new diagnostic required a sensitivity of ≥80% and a specificity of ≥99%.

## Conclusions

The TPPs developed will provide test developers with guidance to ensure that novel diagnostic tests meet identified public health needs.

## Author summary

Accurate diagnostic tests are needed to aid scabies control efforts. In particular they might aid decisions about when to start and stop treatment of whole communities (mass drug administration). Currently most diagnosis is based on clinical examination only and there is a need to establish what criteria new diagnostic tests should meet for them to be of public health use. To aid with this, we determined the programmatic areas of greatest need (use cases) and then developed a shortlist of product requirements (target product profiles, or TPPs) for each scenario. These TPPs can then be used by product developers to ensure that novel diagnostic tools in development are fit for purpose. There were two programmatic use cases for which scabies TPPs were developed. The first TPP focused on diagnostics to determine if the community prevalence of scabies was above 10%—the threshold at which mass drug administration is recommended. The second TPP focused on diagnostics to determine if the community prevalence of scabies is below 2%—the threshold at which mass drug administration can be stopped.

## Introduction

Scabies is caused by infestation with the ectoparasite *Sarcoptes scabiei* var. *hominis* [1]. Infestation leads to itch, skin lesions and in some cases more serious complications due to bacterial superinfection. Scabies occurs worldwide but its distribution is not uniform. In high-income settings, most cases are sporadic. A far higher burden of disease is found in low and middle income countries [2]. In some settings the community prevalence of scabies may be as high as 20–30%. In response to the high burden of disease, individual country commitments to control the disease and the emerging evidence that ivermectin-based mass drug administration (MDA) represents an effective control strategy [3,4], scabies was added to the World Health Organization (WHO) list of Neglected Tropical Diseases (NTDs) in 2017. This was followed in 2019 by an informal consultation, convened by WHO in 2019, and which resulted in a provisional framework for scabies control [5,6].

Central to scale-up of control efforts is the need for more accurate data on the distribution of scabies to guide the roll-out of interventions. Historically, scabies control has been hampered by the lack of a standardised approach to diagnosis, with systematic reviews identifying inconsistency in scabies diagnostic criteria across studies. In response to this the International

Alliance for the Control of Scabies (IACS) used a Delphi consensus method to develop standardized criteria for scabies diagnosis [7,8]. Initial validation studies of these criteria have been conducted in some settings and evaluations of training programmes based on the clinical aspects of these criteria have found that mid-level health care workers can be trained to a standard that ensures an acceptable level of sensitivity, specificity and reproducibility [9–11].

In addition to clinical diagnosis a number of novel diagnostic tests for scabies are being developed [12–14], but none are currently suitable for adoption. Ensuring that diagnostic tools in development are suitable for programmatic use is a critical issue in supporting the scale up of scabies control efforts in line with the 2021–2030 NTD roadmap. At the request of the WHO Department of Control of Neglected Tropical Diseases (NTDs) a single WHO working group was established to identify and prioritize diagnostic needs for each of the 20 NTDs, and to inform WHO strategies and guidance on the subject [15]. The first meeting of this group, known as the Diagnostic Technical Advisory Group (DTAG), was held in Geneva, Switzerland, on 30[th] and 31[st] October 2019. This meeting resulted in the identification of several key priorities for the DTAG. One identified priority was the development of Target Product Profiles (TPPs) for diagnostics to support emerging scabies control programmes.

In this manuscript, we report the process used in developing the TPPs for scabies, the TPP specifications and the assumptions made. The purpose of these TPPs is to support the scabies control strategy in two critical decision areas i) whether the threshold for initiating MDA has been met and ii) whether the threshold for stopping MDA has been met.

## Methods

Based on the recommendation of the DTAG, WHO formed a group of skin-related NTDs to address TPPs for each of these NTDs. A scabies sub-group was established including members of this group and external experts. The scabies subgroup met from November 2020 to April 2021 to agree on priority use cases for the TPPs and undertake the process for the developments of the TPPs. The subgroup leveraged the WHO core TPP development process (Fig 1) as the framework and followed well-established quality planning methodologies [16,17]. Two priority use cases for scabies were agreed upon, based on the recommendations from the recently developed framework for scabies control i) establishing if the proposed community prevalence of scabies for initiating MDA (10%) had been met (starting MDA), and ii) establishing if the proposed threshold for ceasing MDA (2%) had been met (stopping MDA). These recommendations assume that survey will be conducted using cluster randomized, whole age

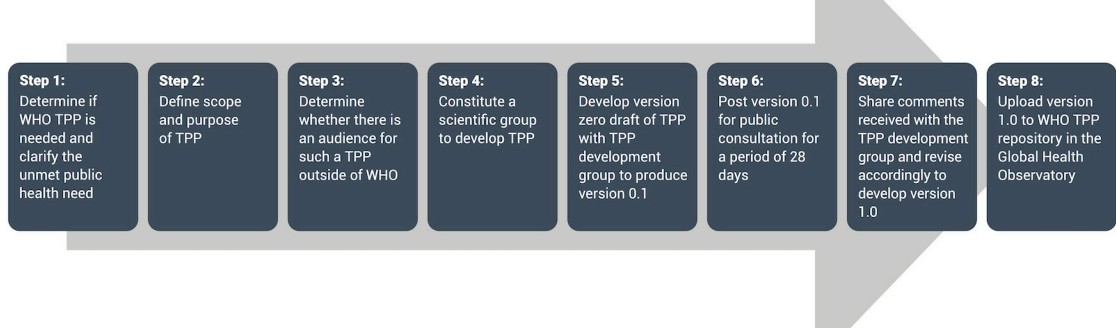

**Fig 1. World Health Organization Target Product Profile Process.**

household survey designs with decisions about MDA being made at the level of the whole evaluation unit. There is currently no agreed design effect for use in scabies surveys so for the purpose of the TPP development we used a design effect of 2. Within the scabies TPP group two expert subgroups were formed, one to determine the attributes required for each scenario (use case characteristics) and another to review the landscape of currently available diagnostic strategies, in addition to a separate modelling of test performance requirements (i.e., sensitivity and specificity) that must be achieved to meet use case objectives. Ultimately, TPPs are intended to facilitate expeditious development of missing diagnostic assays addressing prioritized public health needs. Using the WHO core TPP development process, the expert subgroups for scabies convened online five times to determine the requirements for each use case.

TPPs for each use case considered the following parameters: product use, design, performance, product configuration and cost, and access and equity. Initial 'Draft Zero' requirements in each TPP were selected based on landscape analyses, use case needs analysis and diagnostics performance modeling developed through a consultative process coordinated by WHO Department of the Control of NTDs. For certain elements in each use case, parameters were defined at the outset, and assumptions were made to move forward with sensitivity and specificity calculations. In the context of scabies, the group considered two implementation scenarios. In the first scenario a two-step diagnostic process was considered. This scenario was taken to reflect the process where initial clinical diagnosis was followed by performance of a diagnostic test. In the second scenario a single-step diagnostic process was considered. This scenario was taken to reflect a process whether either clinical diagnosis or performance of a test alone was used as the diagnostic strategy. The committee considered that prevalence surveys would be conducted in line with recommendations arising from the WHO Informal Consultation on Scabies Control. This included the performance of all age, community based cluster randomized surveys. Modelling was used to evaluate the sensitivity and specificity that would be required in each use-case. For the 'Start MDA' use-case sensitivity and specificity were selected such that a true population prevalence of 10% would be detected 80% of the time (false-negative rate 20%) and that a true population prevalence of 8% would only be incorrectly classified as being above the MDA threshold 5% of the time (false-positive rate 5%). For the 'Stop MDA' use-case sensitivity and specificity were selected such that if the true population prevalence were 3% it would be incorrectly classified as <2% only 5% of the time (false-positive rate 5%). In the context of a two-step diagnostic process it was assumed that the sensitivity and specificity of clinical diagnosis were both 80%. It was recognized that this is at the upper limit of the performance reported in evaluations of mid-level healthcare worker diagnosis of scabies[9,10,18,19] and that lower levels of accuracy of initial clinical diagnosis would impact any two-step diagnostic process.

Following development of the initial TPP the scabies subgroup critically reviewed and modified the draft zero where warranted. The draft zero TPP was then sent to the main DTAG committee for review and comments.After revising based on the comments from the DTAG, the scabies subgroup finalized the TPP details, and draft 0.1 TPPs were posted on the WHO website for public comment in July 2021. Comments received were shared with the experts, and TPPs were revised accordingly to generate version 1.0 TPPs.

## Results

The diagnostic landscape review identified clinical and direct identification methodologies as tests that were currently available and antigen and molecular diagnostic tests as in development (Table 1). Estimates of sensitivity and specificity are based on expert opinion informed by the literature.

**Table 1. Diagnostic Landscape of available tests for scabies.**

| Diagnostic tool | Sample type | Analyte | Diagnostic Sensitivity | Diagnostic Specificity | Analytical LOD | Challenges |
|---|---|---|---|---|---|---|
| Physical Examination by an expert | Exposed skin | Visual identification of scabies Best lesions including pathognomic lesions such as burrows | Could be >95% (Considered reference std) | Could be >95%; inter-operator agrement for 2 expert examiners often >95% | Unclear: can diagnose individuals with <10 lesions | Requires highly trained clinicians with many years of experience |
| Physical Examination by a mid-level expert | Exposed skin | Visual identification of common scabies lesions | 65–85% compared to an expert | 65–85% compared to an expert | Varies according to severity of cases. Most programmes have shown reduced sensitivity in milder cases of scabies. | Further evaluation of training packages needed. |
| Dermoscopy | Exposed skin | In vivo mite/ products | Varies between 15–90% depending on expertise and the time allowed for examination | 95–100% when performed by an expert | Single mite | Requires significant specialist training limiting widespread field use |
| Microscopy | Skin scrapings | Mites / Eggs | Dependent on time available—can reach 90% | 95–100% when performed by an expert | Single mite | Significant specialist training is required. Not practical in a field setting. |
| Serology | Serum | Detection of specific antibodies | Currently unknown | Currently unknown | TBD | Cross-reaction with other mites. No well validated assay available. |
| PCR/LAMP | Skin scrapings | Amplified nucleic acid targets | Currently unknown | Currently unknown | TBD | Not a point of care test. High skill level for implementation. Capital costs of laboratory equipment. |

Version 1.0 TPPs for the two use cases were published by WHO on 7 November 2021 within the WHO Global Observatory on Health R&D. No changes were made following the period of public consultation. Select TPP features and their associated requirements are presented in Tables 2 and 3. It was considered a minimum requirement for a test to be lab based and only require standard laboratory equipment. It was considered an ideal requirement that the test could be used at the point-of-care.

**Table 2. Select characteristics of needed test for Starting MDA.**

| Feature | Ideal requirement (Used as a single step-test) | Minimum requirement (Used following an initial screening examination*) |
|---|---|---|
| Intended use | An *in vitro* point-of-care test that detects *S. scabiei*-specific analyte(s) for the purpose of "scabies mapping" to identify areas with ≥10% disease prevalence. | An *in vitro* laboratory-based test that detects *S. scabiei*-specific analyte(s) for the purpose of "scabies mapping" to identify areas with ≥10% disease prevalence. |
| Target analyte | Biomarker(s) specific for current active infection from *S. scabiei*. | Biomarker(s) specific for current active infection from *S. scabiei*. |
| Diagnostic/ clinical sensitivity | 92% | Confirmatory test sensitivity: 96% |
| Diagnostic/ clinical specificity | 98% | Confirmatory test specificity: 84% |
| Cost per test | <USD$1 | <USD$3 |

*Initial clinical screening is considered to have a sensitivity and specificity of 80%.

**Table 3. Select characteristics of needed test for Stopping MDA.**

| Feature | Ideal requirement (Used as a single step-test) | Minimum requirement (Used following an initial screening examination*) |
|---|---|---|
| Intended use | An *in vitro* point-of-care test that detects *S. scabiei*-specific analyte(s) for the purpose of "scabies mapping" to identify areas with ≥10% disease prevalence. | An *in vitro* laboratory based test that detects *S. scabiei*-specific analyte(s) for the purpose of "scabies mapping" to identify areas with ≥10% disease prevalence. |
| Target analyte | Biomarker(s) specific for current active infection from *S. scabiei*. | Biomarker(s) specific for current active infection from *S. scabiei*. |
| Diagnostic/ clinical sensitivity | 80% | Confirmatory test sensitivity: 81% |
| Diagnostic/ clinical specificity | 99% | Confirmatory test specificity: 93% |
| Cost per test | <USD$1 | <USD$3 |

*Initial clinical screening is considered to have a sensitivity and specificity of 80%.

## Discussion

Relative to almost all other NTD programmes global scabies control efforts are at an earlier stage of development. The World Scabies Programme, established in Australia in

2019, represents a first step towards scale-up of scabies control efforts globally, but further work is required to support countries in reaching the ambitious goals for scabies control outlined in the 2021–2030 NTD roadmap [20]. Major challenges facing programmes include the absence of a drug donation programme, the need for further data on the effectiveness of MDA at programmatic scale, and the lack of robust epidemiological data to identify regions where MDA should be rolled-out. Ensuring high-quality, reliable diagnosis of scabies is a critical step in addressing the second of these challenges. The TPP process outlined in this paper represents an important step in establishing criteria for future scabies diagnostics to address programmatic needs.

Currently most activities rely on clinical diagnosis performed by different cadres of healthcare workers. The introduction and validation of the 2020 IACS Consensus criteria for scabies has been an important step in standardizing clinical diagnosis of scabies in research studies [7,9,10], but further work is needed in a programmatic context. In the context of TPP development the group considered two scenarios. In the first scenario performance characteristics were developed assuming a test was performed following initial clinical screening. The sensitivity and specificity of this screening was set at 80% for the purpose of this process. This level of accuracy has been reported in some but not all evaluations of mid-level healthcare worker diagnosis of scabies highlighting the need for further work to improve training for the critical cadre of staff. If the sensitivity of initial clinical examination was substantially below 80% the two-stage process becomes less viable as a strategy, in particular for the decision to initiate MDA. For this reason a single-step test is likely to be preferable if it can be achieved. The single-step TPP criteria require higher levels of performance which surpass those seen in all evaluations of mid-level healthcare workers to date. Other programmes such as trachoma have focused less on achieving particular levels of sensitivity and specificity for mid-level healthcare workers and instead on achieving high levels of inter-operator reliability and setting programmatic thresholds in relation to the accuracy of these mid-level healthcare workers. Whether such approaches could be considered for scabies warrants further consideration.

The provisional framework for scabies control recommends treatment of an entire community predominantly with ivermectin-based MDA in settings where the prevalence of scabies

10% or greater, and that this intervention is continued for between three to five rounds before an assessment of disease prevalence is undertaken. If the community prevalence has fallen to below 2% it is recommended that MDA is ceased, whereas if the prevalence of MDA remains above 2%, further extension of the MDA is recommended [6]. These thresholds are based on current best evidence but may change as further implementation, and operational research studies are undertaken and may be further refined based on insights from mathematical modelling studies [21]. For this TPP, the expert subgroup considered the criteria needed to accurately start MDA at least 80% of the time when the prevalence of scabies was at least 10% and to not incorrectly start if the prevalence was 8% or less more than 5% of the time. The requirement to avoid initiation of MDA reflects, in part, the current absence of evidence about the effectiveness and cost-effectiveness of MDA as a strategy at lower prevalences alongside also the lack of a current drug donation to support MDA progammes, rather than an intrinsic biological rationale for the current threshold. Equally thresholds for stopping reflect expert consensus rather than empirically derived thresholds below which rebound of disease is unlikely following cessation of MDA. The TPP characteristics for both use cases should therefore be considered within these current constraints and the specifications may need to evolve as further data and/or a drug donation or reliable, cheap supply of ivermectin and permethrin becomes available.

## Conclusion

Two TPPs have been presented in this manuscript. The first lays out the specifications for diagnostic tools to detect when MDA should be started and the second when MDA should be ceased. As well as the technical characteristics outlined in this paper and the published TPPS, diagnostic fulfil developed for use in a NTD control or eradication programmes should also be designed to fulfill the WHO reassured criteria. Ideally these TPPs will help inform evaluations of novel diagnostic tests and provide guidance on the standard of clinical diagnosis that would be required if that were to solve as the primary diagnostic modality.

## Acknowledgments

The authors would like to thank all experts and colleagues who provided useful comments through the public consultation.

## Author Contributions

**Conceptualization:** Michael Marks, Christopher Hanna, Kingsley Asiedu.

**Investigation:** Michael Marks.

**Methodology:** Jodie McVernon, James S. McCarthy, Wendemagegn Enbiale, Christopher Hanna, Olivier Chosidow, Daniel Engelman, Andrew Steer.

**Writing – original draft:** Michael Marks, Christopher Hanna.

**Writing – review & editing:** Michael Marks, Jodie McVernon, James S. McCarthy, Wendemagegn Enbiale, Olivier Chosidow, Daniel Engelman, Kingsley Asiedu, Andrew Steer.

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
