## [Decision Letter · Decision Letter 0]

30 Jun 2022

Dear Dr. Marks,

Thank you very much for submitting your manuscript "Diagnostics to support the control of scabies– Development of Two Target Product Profiles" for consideration at PLOS Neglected Tropical Diseases. As with all papers reviewed by the journal, your manuscript was reviewed by members of the editorial board and by several independent reviewers. In light of the reviews (below this email), we would like to invite the resubmission of a significantly-revised version that takes into account the reviewers' comments. 

I would suggest that you replace the letters with the numbers after the authors names and use these numbers (in superscript and adjacent to the first letter of the address) also for the addresses. 

Correct the sentence: i) establishing if the proposed % community prevalence of scabies for initiating MDA (10%) had been met (starting MDA)…

Is there a reason to start with Table 2?> Give explanations what the * in table 2 (1) means

Table 3: Replace the sign ^ with a and continue with b,c,d, instead of ^c, d, and 3!! Why is 3 written in red? Give explanations what all these mean.

Table 4: Give explanations for ^ and 3, replace them with a and b.

Is there any reason why the 3 (whatever it is) is placed in front of the percentages?

Table 3 and 4: Write everywhere S. scabiei. Separate twice Sarcoptesscabiei and replace with S. scabiei (always in italics)

Throughout the text leave a space before the reference numbers.

We cannot make any decision about publication until we have seen the revised manuscript and your response to the reviewers' comments. Your revised manuscript is also likely to be sent to reviewers for further evaluation.

Sincerely,

Kosta Y. Mumcuoglu, PhD

Associate Editor

Jaap van Hellemond

Deputy Editor

I would suggest that you replace the letters with the numbers after the authors names and use these numbers (in superscript and adjacent to the first letter of the address) also for the addresses. 

Correct the sentence: i) establishing if the proposed % community prevalence of scabies for initiating MDA (10%) had been met (starting MDA)…

Is there a reason to start with Table 2?> Give explanations what the * in table 2 (1) means

Table 3: Replace the sign ^ with a and continue with b,c,d, instead of ^c, d, and 3!! Why is 3 written in red? Give explanations what all these mean.

Table 4: Give explanations for ^ and 3, replace them with a and b.

Is there any reason why the 3 (whatever it is) is placed in front of the percentages?

Table 3 and 4: Write everywhere S. scabiei. Separate twice Sarcoptesscabiei and replace with S. scabiei (always in italics)

Throughout the text leave a space before the reference numbers.

Reviewer's Responses to Questions

**Key Review Criteria Required for Acceptance?**

**Methods**

-Are the objectives of the study clearly articulated with a clear testable hypothesis stated?

-Is the study design appropriate to address the stated objectives?

-Is the population clearly described and appropriate for the hypothesis being tested?

-Is the sample size sufficient to ensure adequate power to address the hypothesis being tested?

-Were correct statistical analysis used to support conclusions?

-Are there concerns about ethical or regulatory requirements being met?

Reviewer #1: objectives and rationale clearly stated

Reviewer #2: Authors do a reasonable job describing the TPP process and the basic assumptions used to make the sensitivity and specificity calculations. They should describe the assumptions about the survey design (as cluster design will impact these calculations). It would be helpful to include a table that clearly lays out the various assumptions, as this would help the user understand better how changes in our understanding of the accuracy of the assumptions would impact the TPP

**Results**

-Does the analysis presented match the analysis plan?

-Are the results clearly and completely presented?

-Are the figures (Tables, Images) of sufficient quality for clarity?

Reviewer #1: results well reported

Reviewer #2: The TPPs are presented but other information that is important is missing in this section.

1. It would be helpful for a detailed description of the results of the calculations for sensitivity and specificity (particularly for how the inclusion of the clinical screening prior to the diagnostic test) to be included. Any sensitivity analyses around specific assumptions would be important (e.g. if the performance of the clinical exam is 60%, 70% or 90% how might that impact the needs?

2. In the table about the diagnostic landscape (table 2): are the specificities and sensitivities for the different modalities data-based or expert opinion? Citations should be provided for those numbers that are evidenced-based. If there is a limited amount of evidence, a note to the table indicating that the numbers are expert opinion except when indicated would be important. Also indicate #s from the grey literature.

3. There are a number of footnotes in the tables 3 & 4 that have no corresponding note below the tables (only the ^ appears below the tables. What about 3, c, and d?

4. Very curious as to why the ideal test is LAB-based and not POC. This would be something that would be important to mention in the discussion, as it seems that it would be the other way around.

**Conclusions**

-Are the conclusions supported by the data presented?

-Are the limitations of analysis clearly described?

-Do the authors discuss how these data can be helpful to advance our understanding of the topic under study?

-Is public health relevance addressed?

Reviewer #1: Sharp and to the point

Reviewer #2: Publication of the TPP and a detailed description of how they were developed is a important next step in engaging with those who will help develop the needed tests.

1. Would expound a bit more about the World Scabies Program (and perhaps provide the link); despite its name it is quite small (and underfunded) but it is an important first step towards scale up. A sentence or two describing its structure and funding and clearly stating that it is not a WHO organization would be important.

2. The TPP seems to address the 3 problem challenged mentioned in the 1st paragraph of the discussion, not the second.

3. I was looking for a discussion of how potential changes in the provision thresholds might impact the TPP. Is it likely that the thresholds will go up or down? Which would be more problematic for the TPP?

4. I was also looking for a discussion about how the uncertainty around the calculation of sens/spec of non-expert clinical exam and how that impacts sens/spec calculations for the minimum requirements would be important; should the focus be on a test that does not require exam? Why or why not?

5. If error in a particular assumption could greatly change the needs, some time should be devoted to discussing this. Some would allow for a less stringent profile, others would have the opposite impact. Test developers would need to know this so that they can adjust as the understanding of the true need evolves.

6. What are the implications of not starting MDA (given the 20% risk of not detecting 10% prevalence) vs starting MDA when it not needed (given the 5% risk of finding >10% when the prevalence is 8%)?

**Editorial and Data Presentation Modifications?**

Reviewer #1: (No Response)

Reviewer #2: Numerous typos, run-on sentences, spacing errors, extra periods Genus/species names that are not italicized, etc. Authors should review thoroughly and correct. If future versions of the paper included line numbers it would be easier for review to provide specific feedback.

Recommend table of assumptions, table of sensitivity analyses relevant to the determination of the required test specificity and sensitivity and changes to tables 2, 3, and 4.

If any changes to the TPP were made based on public comment, it would be nice to have that pointed out.

**Summary and General Comments**

Reviewer #1: I have attached some relatively minor edits on a word draft which may be of use

Reviewer #2: The publication of TPP for NTD diagnostics is an important step in the refinement of strategies that will allow progress towards the 2030 goal, and for scabies, for the scale up and demonstration of the impact of control activities. The authors describe the WHO process and many of the assumptions, but much information that would be useful to those would develop new tests is lacking. Adding more detailed description of the calculations, how the 2-step versus 1-step process affects the calculations (and why is a one-step process ideal compared to the 2-step process), how various assumptions impact the TPP, and citations to support some of the specificity and sensitivity claims in table 2 would make this a document that is much more useful to the end users.

PLOS authors have the option to publish the peer review history of their article (what does this mean?). If published, this will include your full peer review and any attached files.

Reviewer #1: Yes: Dr Lucinda Claire Fuller

Reviewer #2: No
---

## [Decision Letter · Decision Letter 1]

8 Aug 2022

Dear Dr. Marks,

We are pleased to inform you that your manuscript 'Diagnostics to support the control of scabies– Development of Two Target Product Profiles' has been provisionally accepted for publication in PLOS Neglected Tropical Diseases.

Please prepare the final version of the manuscript by taking into consideration the remarks of the reviewer below.

Best regards,

Kosta Y. Mumcuoglu, PhD

Academic Editor

Jaap van Hellemond

Section Editor

Reviewer's Responses to Questions

**Key Review Criteria Required for Acceptance?**

**Methods**

-Are the objectives of the study clearly articulated with a clear testable hypothesis stated?

-Is the study design appropriate to address the stated objectives?

-Is the population clearly described and appropriate for the hypothesis being tested?

-Is the sample size sufficient to ensure adequate power to address the hypothesis being tested?

-Were correct statistical analysis used to support conclusions?

-Are there concerns about ethical or regulatory requirements being met?

Reviewer #2: This is a second review. Previous comments are valid.

**Results**

-Does the analysis presented match the analysis plan?

-Are the results clearly and completely presented?

-Are the figures (Tables, Images) of sufficient quality for clarity?

Reviewer #2: This is a second review. Previous comments are valid.

**Conclusions**

-Are the conclusions supported by the data presented?

-Are the limitations of analysis clearly described?

-Do the authors discuss how these data can be helpful to advance our understanding of the topic under study?

-Is public health relevance addressed?

Reviewer #2: This is a second review. Previous comments are valid.

**Editorial and Data Presentation Modifications?**

Reviewer #2: (No Response)

**Summary and General Comments**

Reviewer #2: Thanks for the thoughtful responses and apologies for the confusion about one of my comments.

1. Clarifying my previous comment: Line 207-212. Authors state that the TPP helps address the 2nd problem identified (the need for data on effectiveness of MDA at a programmatic level); I suggest that is would be important both for that issue and for the 3rd challenge regarding the need for robust epidemiological data that indicate where MDA should be rolled out.

2. Ideal vs minimal requirements. I appreciate the additional sentence in the text. However, both table 2 and table 3 still indicate that the IDEAL test is lab-based and the MINIMAL test is point of contact. Please correct the tables.

Minor corrections

1. Line 209, need a comma and not a period

2. Line 205, need a comma before the 'but'

3. Lines 217/218, need a comma before the 'but'

4. Line 224, need a comma before 'then'

5. Line 236, delete comma

6. Line 238, need a comma after 2%

7. Line 238, need either a semicolon or a period before 'whereas'

8. Line 251, need a comma before 'and'

PLOS authors have the option to publish the peer review history of their article (what does this mean?). If published, this will include your full peer review and any attached files.

Reviewer #2: No

---

## [Editor Report · Acceptance letter]

25 Aug 2022

Dear Dr. Marks,

We are delighted to inform you that your manuscript, "Diagnostics to support the control of scabies– Development of Two Target Product Profiles," has been formally accepted for publication in PLOS Neglected Tropical Diseases.

Best regards,

Shaden Kamhawi

co-Editor-in-Chief

Paul Brindley

co-Editor-in-Chief
